# Automatic Inspection of Bridge Bolts Using Unmanned Aerial Vision and Adaptive Scale Unification-Based Deep Learning

**Shang Jiang** [1][ID], **Jian Zhang** [1,*][ID], **Weiguo Wang** [2] **and Yingjun Wang** [3]

1   School of Civil Engineering, Southeast University, Nanjing 211189, China
2   China Railway Construction Suzhou Design & Research Institute, Suzhou 215009, China
3   Guangdong Provincial Communication Group Company Limited, Guangzhou 510623, China
*   Correspondence: jian@seu.edu.cn

**Abstract:** Bolted connections are essential components that require regular inspection to ensure bridge safety. Existing methods mainly rely on traditional artificial vision-based inspection, which is inefficient due to the many bolts of bridges. A vision-based method using deep learning and unmanned aerial vision is proposed to automatically analyze the bridge bolts' condition. The contributions are as follows: (1) Addressing the problems that motion blur often exists in videos captured by unmanned ariel systems (UASs) with high moving speed, and that bolt damage is hard to accurately detect due to the few pixels a single bolt occupies, a bolt image preprocessing method, including image deblurring based on inverse filtering with camera motion model and adaptive scaling based on super-resolution, is proposed to eliminate the motion blur of bolt images and segment them into subimages with uniform bolt size. (2) Addressing the problem that directly applying an object detection network for both bolt detection and classification may lead to the wrong identification of bolt damage, a two-stage detection method is proposed to divide bolt inspection into bolt object segmentation and damage classification. The proposed method was verified on an in-service bridge to detect bolts and classify them into normal bolts, corrosion bolts, and loose bolts. The results show that the proposed method can effectively eliminate the inherent defects of data acquired by UAS and accurately classify the bolt defects, verifying the practicability and high precision of the proposed method.

**Keywords:** bolt inspection; unmanned aerial system; convolutional neural network; motion deblurring; super-resolution

## 1. Introduction

Bolt connections are a commonly used connection type in steel bridges. The bolt connection not only needs to bear the enormous load caused by the bridge's weight, traffic load, and temperature, but also is affected by the continuous vibration of the bridge caused by the wind load and the dynamic load of the vehicles. Therefore, bolt damage in steel bridges is one of the representative failure forms. It generally includes bolt corrosion, bolt looseness, or even missing bolts. Once a bolt is loose or missing, it means that the bearing capacity of the connection has been seriously degraded. Therefore, it is of great significance to inspect bridge bolts regularly. The existing bridge bolt detection methods generally rely on manual visual inspection, which is inefficient and dangerous, especially for long-span bridge inspection. For example, a long-span steel truss bridge usually contains hundreds of thousands of bolts. Manual vision inspection is time-consuming. To detect cable clamp bolts of a suspension bridge across a river, workers must climb to a cable more than 200 m high to observe the cable clamp bolts one by one, which is quite dangerous.

The existing contact-based bolt looseness detection methods (including acoustic/ultrasonic methods, impact testing, the vibration-based approach, and piezoelectric-enabled active sensing [1–4]) have played an essential role in inspecting steel structure bolts. These

methods can quantitatively analyze bolt looseness and even measure the axial force of bolts. They are of great significance for the monitoring of bolts' condition in essential parts. However, more efficient methods need to be adopted for the daily rapid inspection of massive bolts on long-span bridges. In recent years, a more widely used image-based visual inspection method has made feasible the rapid and automatic assessment of bridge bolts. This kind of non-contact detection method uses a camera to collect bolt images rapidly and uses analysis methods based on machine vision to automatically identify bolts and classify whether there are defects in bolts. It is more efficient than the contact-based bolt inspection method and avoids the problem of contact detection being sensitive to environmental noise and relying on strict operations. Park et al. [5] proposed a bolt detection method using edge detection. Then, Hough transform was proposed to calculate the rotation of bolts so that bolt loosening could be determined by comparing the rotation angle of the bolt in the two periods. A similar method was proposed by Cha et al. [6], who added a linear support vector machine to distinguish tight bolts from loose bolts so that results could be more robust. However, the method based on Hough transform can only calculate bolt rotation of 0~60 degrees and needs to take photos of the bolt in different periods, which is not suitable for the rapid inspection of a large number of bridge bolts. Ramana et al. [7] applied the Viola–Jones algorithm, which can identify loose bolts automatically, and the reported accuracy on an indoor test was 97%. Sun et al. [8] proposed a fast bolt-loosening detection method using a convolutional neural network (CNN) and binocular vision. This method can quantitatively judge whether the bolt is loose by calculating the moving distance of the bolt. These methods are mainly based on image processing (IP), but IP methods are easily affected by the background of the bolt image and uneven illumination.

Recently developed deep learning methods have been proven to recognize features from complex background images [9–11], and which have higher stability and accuracy than traditional IP methods, have begun to play an increasingly important role in structural damage detection. Hu et al. [12] proposed a network named the Deep Automatic Building Extraction Network (DABE-Net) and adopted it to extract buildings from remote sensing imagery. An automated crack detection technology based on an improved YOLO v4 network was put forward by Yu et al. [13], and the reported mean average precision was 0.976. The VGGnet architecture was applied by Davis et al. [14] to classify construction waste material automatically, and the delivered accuracy was 94%. Dong et al. [15] reviewed the application of computer vision-based methods on structural health monitoring. The application scenarios included structural damage detection (including cracks, concrete spalling, steel structure corrosion, and bolt looseness), 3D reconstruction, and so on [16,17]. The method of bolt looseness detection using deep learning methods has also been studied in recent years. A Single Shot Multi-Box Detector (SSD) was applied by Zhao et al. [18], and the evaluated accuracy on images obtained from different angles and lighting conditions was 0.914. The region-based convolutional neural network R-CNN model was used by Pham et al. [19] to detect bolt loosening. This method was evaluated on the real-scale bolted connections of a historical truss bridge and showed high application potential. Similarly, state-of-the-art networks, including RCNN [20], faster RCNN [21], and mask RCNN [22], have been applied to bolt detection, too. These studies showed that using deep learning methods to quickly and automatically detect loose bolts is expected to be applied to the health maintenance of in-service bridges. Most existing research uses laboratory components as the dataset collection objects and in verification experiments. However, the number of images in these datasets is generally less than 300, and many of them are images with repetition rates. The bolts on the components in the laboratory are also different from those on in-service bridges.

The image acquisition of bridge bolts is also difficult for traditional inspection methods, while existing research has hardly been studied. The method of using UAS loaded with a camera is expected to solve this problem [23–26]. With the maturity of flight control systems for UASs and the decline in prices, more and more UASs have been used in bridge inspections due to easy operation and low cost. Chen et al. [27] proposed a UAS-based

method that used homographic transformation and digital image correlation to measure structural vibration. A wall-climbing UAS, designed by Jiang et al. [28] and applied to structure crack inspection, can switch between flying and wall-climbing modes. A UAS with an ultrasonic beacon system was suggested by Kang et al. [25], and the UAS can fly along the bridge bottom while inspecting concrete cracks. Although research on bridge inspection using UASs has been launched, using UASs for bridge bolt inspection has received no attention.

It is important to note that among the existing research on using UASs to detect bridge damage, there is little research on ensuring the quality of image data collected by UASs. When the UAS is flying at high speed, there is likely motion blur in the image collected. Meanwhile, due to the frequent changes in distance between the camera and structure during flight, how the feature scale in the captured image changes and how to deal with images with multi-scale features is also a task.

To address these problems, a practical method that uses images collected by UAS and a deep learning-based automatic analysis method are proposed to inspect bolts from an in-service bridge with high precision and high efficiency. What is different from existing research is that the proposed method aims to develop strategies that use UAS to collect bridge bolt images, reduce the quality deterioration of images caused by high-speed flight, detect bolts, and classify bolt damage with high accuracy. Specifically, a UAS designed for bridge bolt inspection was designed and modified; then, the method of using this UAS to collect bridge bolts was proposed. Second, a method based on optical flow and inverse filtering was proposed to determine whether there was motion blur in UAS video and automatically eliminate motion blur. Then, the image was segmented by adopting a proposed super-resolution-based adaptive scaling method into subimages with uniform bolt sizes. After that, bolt damage was determined with a two-stage bolt inspection method, in which the YOLO v5x network trained on a custom dataset was applied to detect bolts and segment them into single bolt images. Then, efficientNet was used to classify bolts into normal bolts, corrosion bolts, and loose bolts.

The remainder of this paper is organized as follows: the framework of the proposed bolt inspection method is introduced in Section 2. The designed UAS, bolt data acquisition method using the UAS, and bolt image preprocessing method are presented in Section 3. Section 4 discusses the proposed two-stage bolt inspection method using deep learning. The field test is shown in Section 5, followed by the conclusions in Section 6.

## 2. Framework of the Proposed Method

The proposed automatic bridge bolt inspection method is a vision-based method using image processing and deep learning algorithms with images acquired by UAS, which contains three parts: bolt image acquisition, data preprocessing, and bolt damage identification, as shown in Figure 1. In detail, bolt image acquisition used the modified UAS to capture the bolts at the bridge's side, bottom, and cable clamp according to the specific route. Data preprocessing aims at two problems that may cause image degradation when using UAS: the first is image motion blur caused by the high moving speed of the UAS, which causes image motion blur while taking video or image. The other is that different object distances lead to large changes in the pixel size of the bolts in images. Usually, a single bolt occupies a small size in the image when the distance is large. These problems may lead to low accuracy and automatic bolt damage identification instability. To address these problems, an image motion deblurring method using inverse filtering with a camera motion model calculated by optical flow was proposed to estimate and eliminate motion blur automatically; then, an adaptive scale segmentation method using multi-scale template matching and super-resolution based on Enhanced Super-Resolution Generative Adversarial Networks (ESRGAN) was proposed to segment bolt images into uniform scale subimages with a bolt pixel size of 240 × 240 pixels. The third part was to use a proposed two-stage bolt damage identification method to detect bolts in images and classify the damage types, in which the YOLO v5 network was applied to detect and segment bolts,

then efficientNet was used to classify the segmented bolts into normal bolts, corroded bolts, and loose bolts.

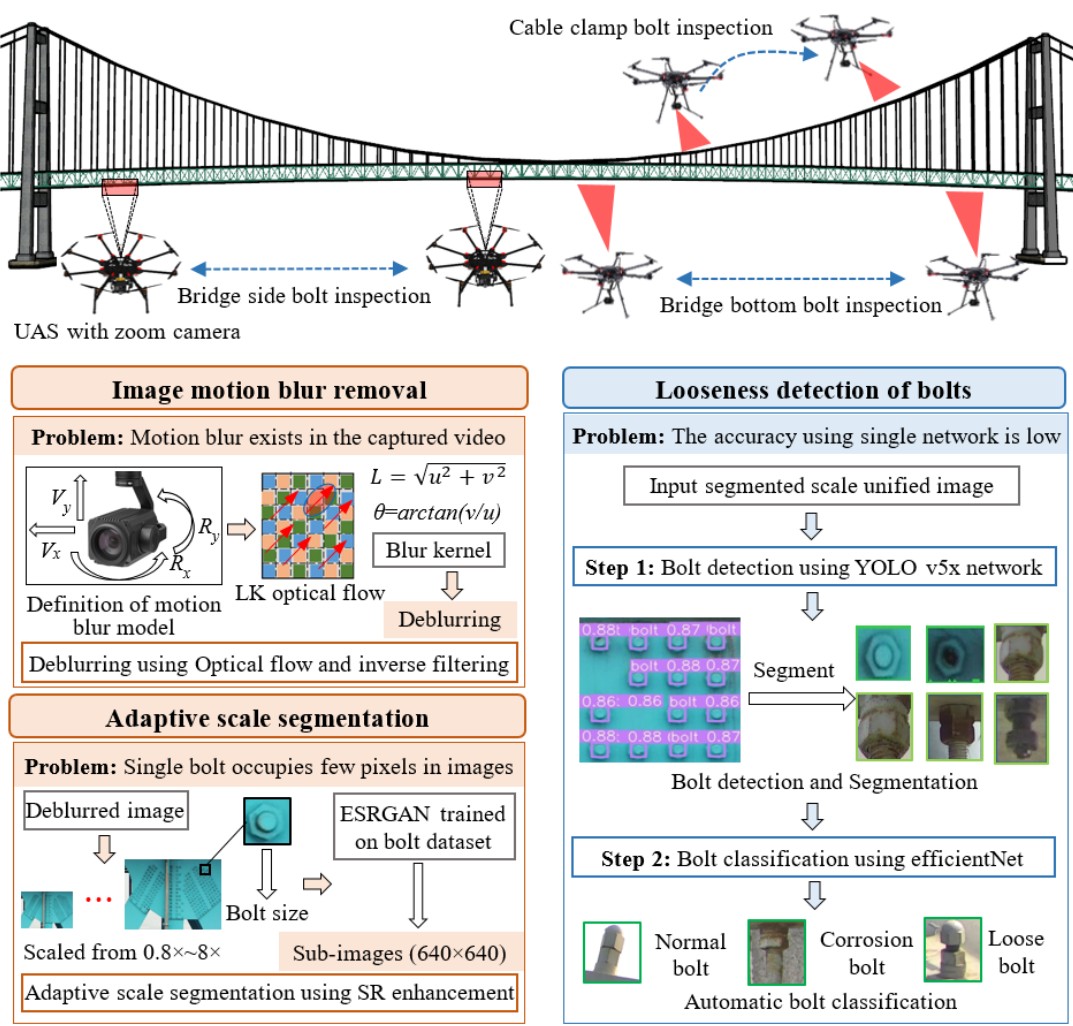

**Figure 1.** The framework of the proposed method.

## 3. Data Acquisition and Preprocessing

### 3.1. Design of the UAS

UASs have been widely applied in inspecting structural damage, including cracks, concrete spalling, etc. When selecting a UAS for bridge bolt inspection, the endurance time, positioning accuracy, camera focal length, and computing ability should be considered comprehensively. The designed UAS is shown in Figure 2, in which matrice-600pro from DJI Company was used as the UAS platform, the zenmuse-Z30 camera was used to acquire bolt images or videos, and the Intel NUC 10 micro-computer was used for onboard computing. The total weight of the UAS was 11 kg and the endurance time was about 40 min. The Z30 camera was modified to be upside down so that the UAS could take bolt images at the bottom of the bridge. The UAS was equipped with three sets of GPS antennas so that the received GPS signals could be more stable than those of a traditional UAS. Meanwhile, the Z30 camera was a zooming camera with a maximum of 872 mm of 35 mm equivalent focal length, so that clear bolt images can be obtained without coming close to the bridge surface. The Intel NUC 10 micro-computer could read video from the Z30 camera directly through the onboard software development kit (OSDK) and perform continuous data processing ability when algorithms were transplanted.

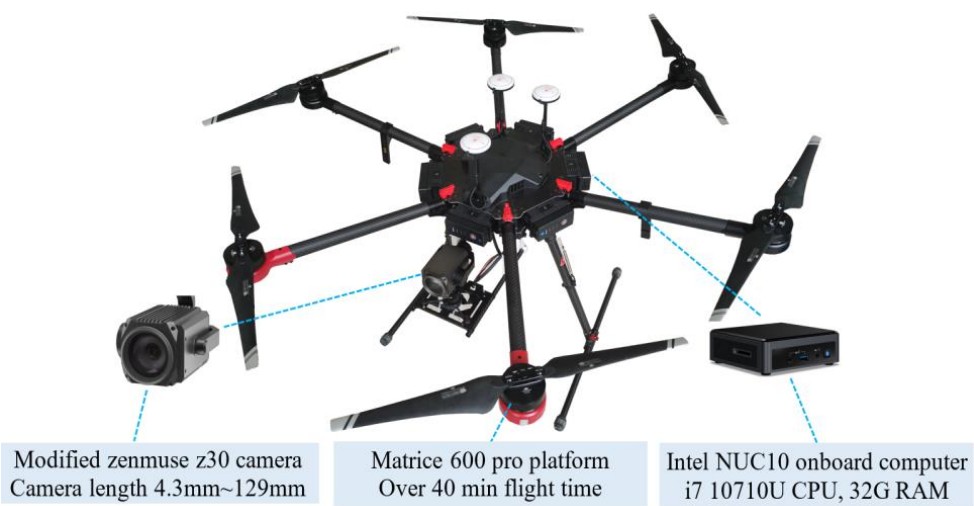

Modified zenmuse z30 camera
Camera length 4.3mm~129mm

Matrice 600 pro platform
Over 40 min flight time

Intel NUC10 onboard computer
i7 10710U CPU, 32G RAM

**Figure 2.** Design of the UAS.

### 3.2. Strategy for Bridge Bolt Data Acquisition Using UAS

Despite the advantage that traffic would not be affected when using a UAS based inspection method, safety should be considered first when determining the bolt image acquisition strategy due to the severe consequences of any accident. Therefore, an appropriate way is to keep a safe distance between the UAS and the bridge and use the long focal length of the zoom camera to acquire images. Three flight paths and photography strategies were designed corresponding to the position of the bridge where the bolts were located. (1) Flight paths for side bolt inspection were several routes parallel to the side of the bridge according to the direction of the bridge. Taking a steel truss bridge as an example, the bolt connection joints of the bridge side are located in the upper, middle, and bottom lines of the vertical side of the bridge. Therefore, the side routes are three routes with plane overlap, and elevations are the upper, middle, and bottom elevations of the bridge side, respectively. (2) Flight paths for bolts on cable clamp are trajectories paralleling the bridge cable, considering that it is hard to control a UAS to fly in parallel with a bridge cable because the cable is nonlinear. The strategy is to hover the UAS at different positions on the side of the cable and rotate the camera to capture the clamp images. (3) For bolt inspection at the bridge bottom, a similar strategy to cable clamp inspection is used due to the complex environment at the bridge bottom. Photography strategies are shown in Figure 1, which shows the distance between UAS and the bridge is 20 m when capturing side bolts, and it is 10 m when capturing bolts at the bridge bottom. Because UASs have a short flight period, achieving high detection efficiency is one of the most critical aspects of the UAS-based bridge inspection approach. If the bolt image is acquired by taking photographs, the UAS will stop, hover, and adjust the angle to take photos at each bolt connection portion, considerably increasing the complexity of operation and flying duration. As a result, the bolt data is more efficiently gathered by recording videos. During inspection, the UAS can continue to fly or operate the camera. Furthermore, the method of taking continuous video means it is more difficult to miss capturing targets than when taking images.

### 3.3. Zoom Camera Model and Motion Deblurring

The most widely used camera model is the pinhole model. It has been used in many existing works [29]. The camera equipped on the UAS was a zoom camera, so the focal length $f$ was a variable based on the pinhole model, as shown in Figure 3a. Based on the pinhole model, if the measured object has a displacement at the moment of image capture that is, the moment when the electronic shutter controls the charge-coupled element to produce photoelectric induction in the vertical direction of the photosensitive plate,

compared with the object pixel obtained by the first line of photosensitive material, the object pixel displacement obtained by the Nth line of photosensitive material is:

$$\Delta x_p = \frac{vt}{f + \Delta f} \tag{1}$$

where $v$ is the horizontal velocity of UAS, $t$ is the exposure time of each frame. For a video with an acquisition frequency of 30 fps, $t = 0.033$ s. $f + \Delta f$ is the focal length of the camera, as shown in Figure 3b.

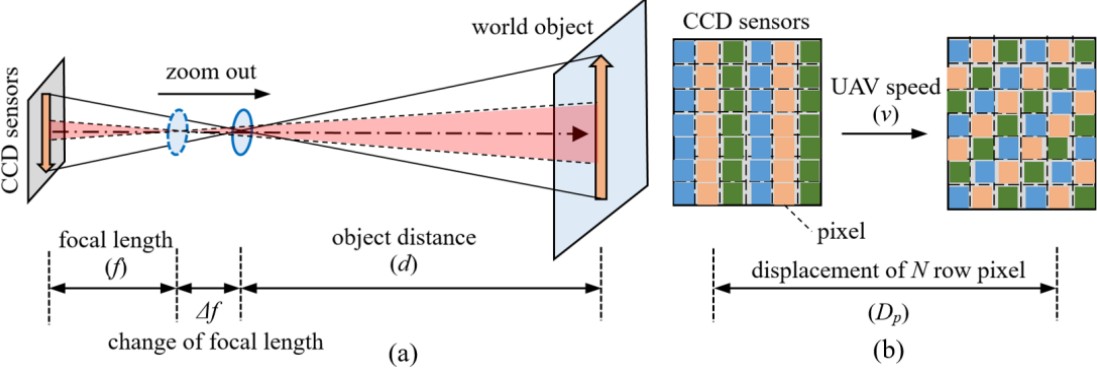

**Figure 3.** Zoom camera model (**a**) and motion blur principle (**b**).

The degraded image $g(x, y)$ caused by camera movement and noise during UAS-based imaging can be considered as the convolution of the interference-free image $f(x, y)$ and the point spread function $p(x, y)$ plus the environment noise $n(x, y)$, expressed as:

$$g(x, y) = f(x, y) \otimes p(x, y) + n(x, y) \tag{2}$$

where $x$ and $y$ represent the horizontal and vertical coordinates of pixels in the image coordinate system. For an image with size *[M, N]*, the pixel in it can be expressed as $p_{i,j}(x_i, y_j)$, where $0 \le i \le M, 0 \le j \le N$, and the image composed of $p_{ij}(x_i, y_j)$ can be expressed as $f(x, y)$. Since the speed of the UAS did not change significantly during flight and the exposure time of each frame was less than 10 ms, within this time, the displacement of the UAS was minimal. The relative displacement of the camera and the photographed bridge surface can be considered a uniform linear motion. In addition, inspection is usually conducted in good weather with enough light, so the noise caused by light and other conditions can also be ignored. Under this uniform linear motion, the exposure during image shooting can be expressed as:

$$\begin{cases} \left(\frac{\partial}{\partial t} + v\frac{\partial}{\partial x}\right)e(x, t) = 0 \\ e(x, 0) = f(x) \end{cases} \tag{3}$$

where $e(x, t)$ is the exposure of the camera at time $t$; when $t=0$, it is expressed as *f(x)*, then it can be defined as $e(x,t) = f(x - vt)$ at any time $t$. The image acquired on the image sensor at this time is the cumulative exposure at that moment, expressed as:

$$g(x, t) = \int_0^t e(x, t)dt = \int_0^t f(x - vt)dt \tag{4}$$

If $\varepsilon = x - vt$, the above formula can be transformed into:

$$g(x, t) = \frac{1}{v} \int_{x-vt}^{x} f(\varepsilon)d\varepsilon \tag{5}$$

The above formula shows that the motion blur is the result of the superposition of multiple images in the $[x - vt, x]$ interval at $t = 0$. If the UAS is hovering, namely $v = 0$, then

$$g(x,t) = \int_0^t f(x)dt = tf(x) \tag{6}$$

This means that the camera is continuously taking the same image. Further, if the UAS is not moving in a horizontal direction, but flying at a constant speed $v$ at any angle $\theta$, then

$$g(x,y) = \int_0^t f(x - v_p t cos\theta, y - v_p t sin\theta)dt \tag{7}$$

where $v_p$ is the pixel speed converted from the camera model and the actual flying speed $v$. It can be seen that the direction of motion $\theta$ and velocity $v_p$ are the two key parameters to determine the motion-blurred image.

Since the Z30 zoom camera was used in the inspection, the camera parameters needed to be set according to the lighting conditions when taking photos. During the inspection, the exposure time was set to $t_e =$ 1/100 s, and the resolution was $w \times h$ = 1920 × 1080. Therefore, in the exposure time of each frame, the sensing speed of the photoelectric sensor was $v_e = \frac{h}{t_e} = 1.08 \times 10^5$ pixel/s. Generally, if the motion of blurred pixels reaches 6 pixels, the motion blur of the image can be felt and may cause image degradation. So that the relative pixel motion speed $V$ between the camera and the bridge should not exceed $v_p = \frac{6}{t_e} = 600$ pixel/s, the pixel movement of a single frame relative to the previous frame should not exceed 20 pixels when the UAS takes video at 30 fps. This can be used to judge whether motion blur occurs in bolt videos.

After that, it is vital to determine the direction of the motion blur and the size of the blur kernel. For the determination of the motion blur direction $\theta$, it is known from the photography strategies in Section 3.1 that for bridge side bolt inspection, the UAS was moving horizontally to the bridge, so the motion blur direction should be 0° or 180°. For the bolt inspection at the bridge bottom, the UAS hovers with the camera rotating horizontally. Since the camera rotates at a slight angle at every moment, the movement direction $\theta$ can be considered to be 90° or 270°. For bolt inspection in the cable clamp, the camera basically rotates horizontally, so the motion blur direction is similar to that in the side bolt inspection.

For the determination of the blur kernel size, the key is to determine the amount of motion of the pixels in the image at each moment. When the amount of movement is greater than 20 pixels, the motion blur of the image needs to be eliminated, and the size of the blur kernel is the calculated pixel displacement.

The above problem can be summarized as the need to evaluate which frames in the captured video need to take the role of motion blur elimination and calculate the blur direction and blur kernel size. The problem is then summarized as the calculation of displacement and vector direction between adjacent frames in the video. An image motion blur detection and deblurring method based on optical flow and inverse filtering was proposed to address this problem. First, the Lucas–Kanade (LK) optical flow algorithm [30] was used to automatically calculate the displacement and direction of adjacent frames so blurred frames can be selected automatically, and the displacement and direction are the sizes of the blur kernel and the direction of motion blur. The inverse filtering method was used with the calculated blur kernel size and direction to remove the motion blur directly. It is calculated as shown in Equation (8), where $T$ is the exposure time, and a and b are the horizontal and vertical movement. The steps of the process are shown in Figure 4.

$$H(u,v) = \frac{Tsin[\pi(ua + vb)]}{\pi(ua + vb)}e^{-j\pi(ua+vb)} \tag{8}$$

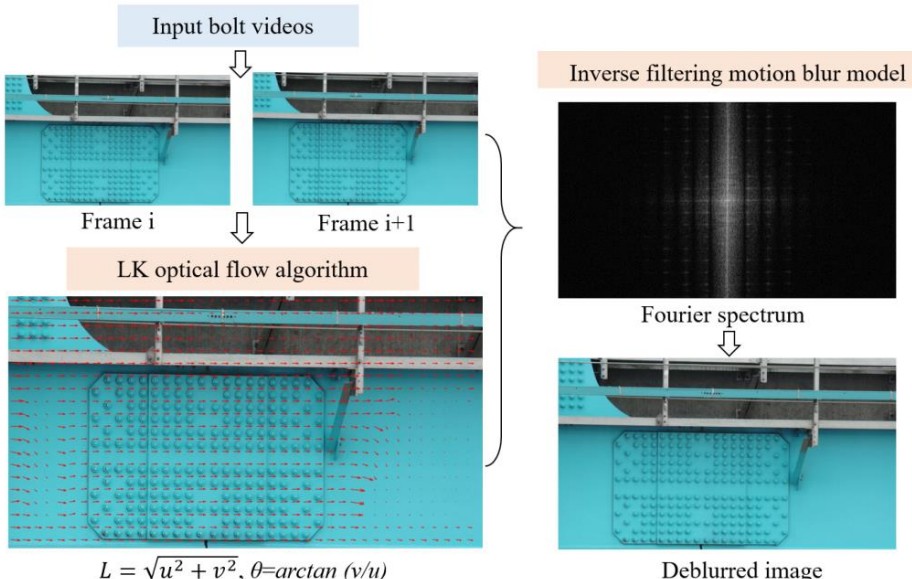

**Figure 4.** Motion deblurring using optical flow and inverse filtering.

### 3.4. Adaptive Scale Segmentation Based on ESRGAN

The purpose of adaptive scale segmentation is to unify the size of single bolts after bolt detection, and enhance the details as much as possible, so the classification accuracy of bolt loss can be stable and accurate. The core of the proposed adaptive scale segmentation method is to enlarge the bolt images taken at different distances so that the scale of a single bolt in the scaled bolt image will be close to the input size of the classification model, which is 240 × 240 pixels. The steps to achieve this goal were as follows: First, a multi-scale template matching was used to find the circumscribed rectangle size of the smallest bolt in the image acquired by the UAS. The ratio of the long side of the circumscribed rectangle to 240 pixels is the enlarged ratio of the raw image. Then, rounding the enlarge ratio and comparing it with two times, four times, and eight times, the closest one was the final enlarge ratio of the raw image. Then, the pre-trained ESRGAN network was used to enlarge the source image twice, four times, or eight times. Finally, a sliding window method was used to segment the enlarged image with a 20% overlap rate. After that, the captured image would become the subimage with a uniform scale. The operation steps of the method are shown in Figure 5.

The first step of the proposed method was to confirm bolt sizes from raw images. The template matching method was adopted considering the timeliness of the algorithm. Several standard bolt images were set first and then we traversed the collected images to find the area with the highest matching degree with the raw image. The size of the highest matching area obtained was the size of the bolt in the image. Considering that traditional template matching is not suitable for image matching with multi-scale, a multi-scale transformation was added before template matching so that the optimal scale of the bolt size could be determined adaptively by choosing the optimal matching. The specific steps of the algorithm were as follows: firstly, the preset image was scaled from 0.8 to 8 times, and template matching was carried out under each scaling factor. The applied matching method was the sum of squared differences, which can be expressed as:

$$D(i,j) = \sum_{s=1}^{M} \sum_{t=1}^{N} [S(i+s-1, j+t-1) - T(s,t)]^2 \tag{9}$$

where *S(s,t)* is the search image of size $M \times N$, $T(s,t)$ is the template image of size $m \times n$, and *(i,j)* is the coordinate of the upper left corner in the process of traversal searching the image. After traversing, the maximum matching area of $D(i,j)$ was selected as the bolt position.

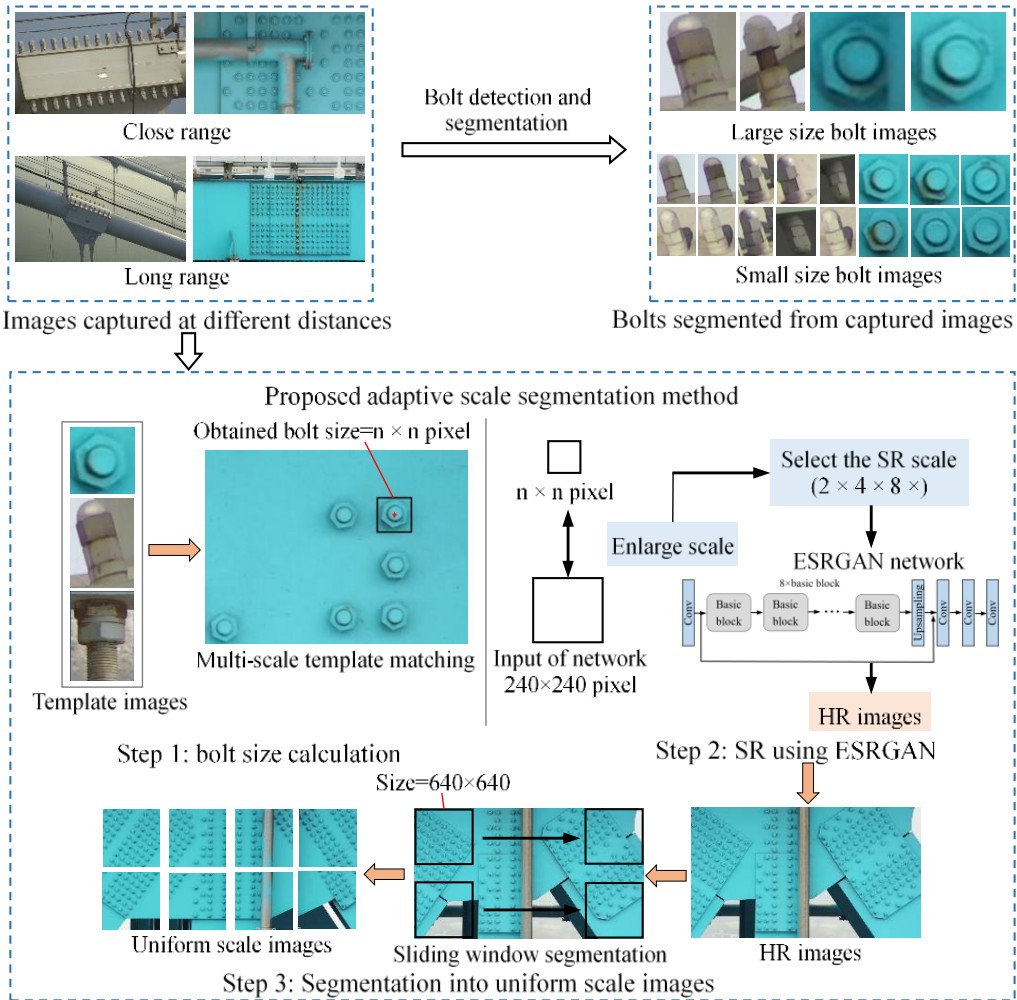

**Figure 5.** Workflow of adaptive scale segmentation method.

Once the bolt size was obtained by template matching, it was compared with the input size of the classification network (240 × 240 pixel) to be a multiple of the super-resolution required for the image. Since the trained super-resolution network generally only had a magnification ratio of 2, 4, and 8 times, it was necessary to round the calculated multiple. Image or video super-resolution is one of the early applications of deep learning. The purpose of super-resolution is to recover a large-size high-resolution image from a small-size low-resolution image so that the image retains more details. The super-resolution method based on deep learning is generally based on single image super-resolution (SISR), and the network is used to learn the end-to-end mapping function from low-resolution images to high-resolution images in several pairs of low-high-resolution images in the training set so that it can handle super-resolution tasks similar to the images in the training set.

Research on image super-resolution using deep learning has been done in recent years. The super-resolution convolutional neural network (SRCNN) was the earliest one to use bicubic interpolation to enlarge images and then perform nonlinear mapping [31] through a three-layer convolutional network to obtain a high-resolution image result [32], but the layers of a SRCNN are few, and the receptive field is small (13 × 13). Then, the deeply-recursive convolutional network (DRCN) with more convolution layers and greater receptive fields was proposed [33], which also uses a recurrent neural network (RNN) to avoid too many network parameters. In response to the low efficiency of DRCN, an efficient sub-pixel convolutional neural network (ESPCN) using the sub-pixel convolution operation to directly perform the convolution operation on the low-resolution image was

raised, which can significantly improve the inference efficiency [34]. A super-resolution generative adversarial network (SRGAN) is a super-resolution network that uses a generative adversarial network (GAN) as a discriminator [35]. It overcomes the problem of using mean square error (MSE) as a loss function in previous networks, which causes the generated image to be too smooth and lack high-frequency texture details. Its loss function is composed of adversarial loss and content loss so that the quality of the generated image is not judged from the pixel level but from the high-level abstract features. An enhanced super-resolution generative adversarial network (ESRGAN) is an improved network to SRGAN [36]. Aiming at the problem of unrealistic texture details and accompanying noise generated by the original model, the original model's network structure, counter loss function, and perceptual loss function were respectively improved, specifically using the residual-in-residual dense block (RRDB) method instead of residual blocks. By deleting the batch normalization (BN) layer to prevent the introduction of irrelevant features in training, a relative generative adversarial network (relativistic GAN) was used to let the discriminator predict relative realism instead of absolute value, and the former activated feature expression with stronger supervision information was used to constrain the perceptual loss function.

Secondly, the ESRGAN was used as the super-resolution network, and the network structure is shown in Figure 6. The characteristics of bolt images were entirely different from those in the existing super-resolution open-source database, so it was necessary to establish a super-resolution database of bolts. Considering that the data must be as close as possible to the in-service bridges, we used the modified UAS to collect 31,087 bolt images from three in-service bridges. Five hundred original images were selected from the data of each bridge as high-resolution (HR) images in the training set, and we resized these 1500 HR images by 0.5, 0.25, and 0.125 times to make the two, four, and eight times super-resolution low-resolution (LR) images respectively using the resize function of Matlab. Based on this, a super-resolution database of bridge bolts was made.

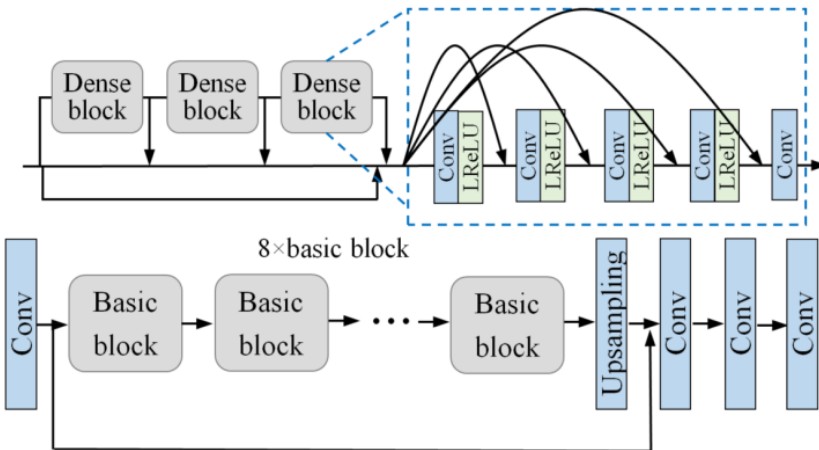

**Figure 6.** Structure of ESRGAN.

During training, the result of the network should be evaluated, and the most commonly used paraments, including peak signal-to-noise ratio (PSNR), structural similarity index (SSIM), and Laplace gradient sum, are used as the evaluation parameters. PSNR is the ratio of the maximum power of the signal to the noise power of the signal, which was used to measure the quality of the compressed reconstructed images. It was calculated as:

$$PSNR = 10 \times log_{10}\left(\frac{MAX_I^2}{MSE}\right) \tag{10}$$

where $MAX_I$ represents the maximum value of the pixel in the image, and mean square error (MSE) represents the mean squared value of two corresponding images [37]. The MSE of two images $I(i,j)$ and $T(i,j)$ with size $M \times N$ can be calculated as:

$$\text{MSE} = \frac{1}{M \times N} \sum_{i=1}^{N} \sum_{j=1}^{M} \left( I_{ij} - T_{ij} \right)^2 \tag{11}$$

SSIM represents the degree of image distortion and its calculated method as:

$$\text{SSIM}(I, T) = \frac{(2u_I u_T + C_1)(2\sigma_{IT} + C_2)}{\left( u_I^2 + u_T^2 + C_1 \right)\left( \sigma_I^2 + \sigma_T^2 + C_2 \right)} \tag{12}$$

where $u_I$ and $u_T$ are the mean values of images $I(i,j)$ and $T(i,j)$, $\sigma_I$ and $\sigma_T$ are the standard deviations of the two images, $\sigma_I^2$ and $\sigma_T^2$ are the variances of the two images, respectively, and $\sigma_{IT}$ is the covariance of the two images, while $C_1$ and $C_2$ are constants. All of the above were calculated using the Gaussian function [38].

The Laplace gradient sum is a common index to evaluate image clarity. Its calculation method is to obtain the Laplace gradient value of pixels with the Laplace template and calculate the sum of the Laplace gradient values of all pixels. The applied Laplace template is:

$$\text{s} = \sum_{x=2}^{N_x-1} \sum_{y=2}^{N_y-1} |f(x+1,y) + f(x-1,y) + f(x,y+1) + f(x,y-1) - 4f(x,y)| \tag{13}$$

where $N_x$ and $N_y$ are the number of horizontal pixels and vertical pixels of the image. The Laplace gradient sum in Table 1 is the average gradient sum of the selected 100 images.

**Table 1.** Comparison of super-resolution methods.

| Evaluating Indicator | Times | Bicubic Interpolation | VDSR | ESRGAN |
|---|---|---|---|---|
| | 2× | 36.03 | 38.00 | 38.83 |
| PSNR | 4× | 31.21 | 33.30 | 33.93 |
| | 8× | 27.63 | 29.13 | 29.70 |
| | 2× | 0.95 | 0.97 | 0.97 |
| SSIM | 4× | 0.93 | 0.94 | 0.94 |
| | 8× | 0.87 | 0.90 | 0.91 |
| | 2× | 20.30 | 50.73 | 52.75 |
| Laplace gradient sum | 4× | 4.71 | 15.51 | 51.47 |
| | 8× | 2.91 | 9.83 | 24.49 |

The model was trained on TensorFlow, where the generator network of ESRGAN was a residual-in-residual dense block net (RDDBNet), and the VGG-19 network was used as the discriminator network. The number of training steps was 50,000, and the learning rate was 0.0001. After training, 100 images were randomly selected from the bolt images of the bridge to calculate the PSNR and SSIM between 2×, 4×, and 8× super-resolution images enlarged using ESRGAN and original images. Other super-resolution methods, including bicubic interpolation and the VDSR trained under the same conditions, were used as the comparison, and the results are shown in the Table 1:

The results show that the bridge bolt image enlarged using ESRGAN show better image quality at 2×, 4×, and 8× than the other two methods. Figure 7 is a comparison of images enlarged using three methods. Since two times and four times super-resolution were the most used, the comparison is the effect of 4 times super-resolution. It can be seen that the ESRGAN showed a higher degree of restoration in the super-resolution restoration of bolt images, especially in the restoration of bolt edges, thread textures, and corrosion

textures. This advantage provides stable and reliable source data for bolt identification and classification in the following processing.

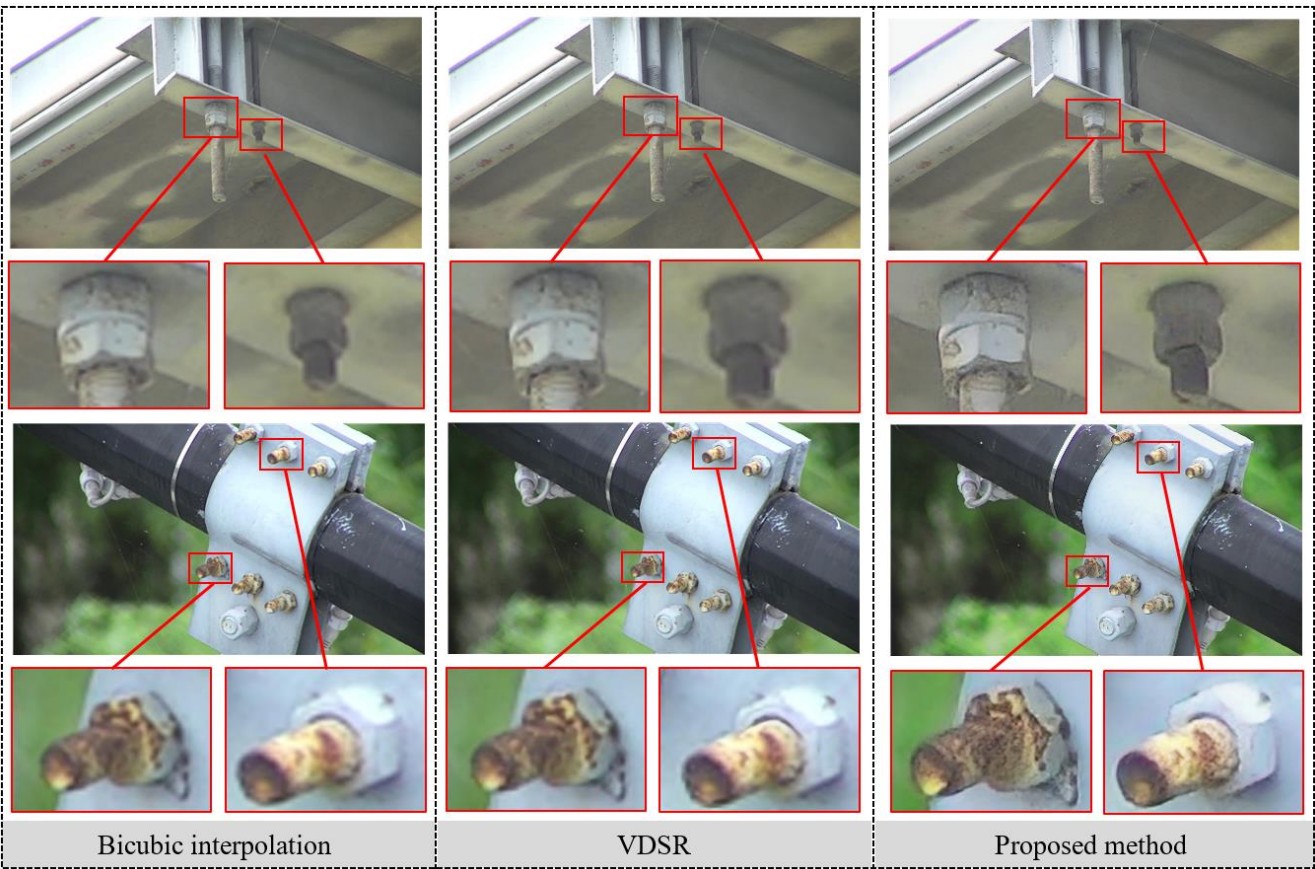

**Figure 7.** Comparison of super-resolution results using bicubic interpolation (**left**), VDSR (**middle**), and the proposed method (**right**).

## 4. Two-Stage Bolt Inspection Based on Deep Learning

Deep learning methods have been increasingly applied in various fields of structure inspection in recent years. State-of-the-art networks are not only close to or even better than manual inspection in the recognition accuracy of complex images but also have shown significant improvements over previous networks in terms of the increase in inference speed and the reduction of computational consumption. Among them, the YOLO series networks are the best networks as well as the most widely used networks [39]. YOLO v5 is the latest network in the YOLO series. The first version of YOLO v5 was proposed in June 2020. Considering that the latest YOLO v5 series networks already have a high inference speed and detection accuracy, verification of a large number of databases has shown that it meets engineering needs. The YOLO v5 series networks have strong multi-platform portability, so the latest YOLO v5 series networks were applied, and the contributions were focused on selecting the optimal model from the four models of YOLO v5s, YOLO v5m, YOLO v5l, and YOLO v5x, and customizing the model parameters on the established in-service bridge bolt database.

Figure 8 shows the network structure of YOLO v5s. Other network structures of the YOLO v5 series are similar, except that the cross-stage partial (CSP) structure with different numbers of residual components and the focus structure with different numbers of convolution kernels are used. The network could be divided into four parts: the input, backbone, neck, and prediction.

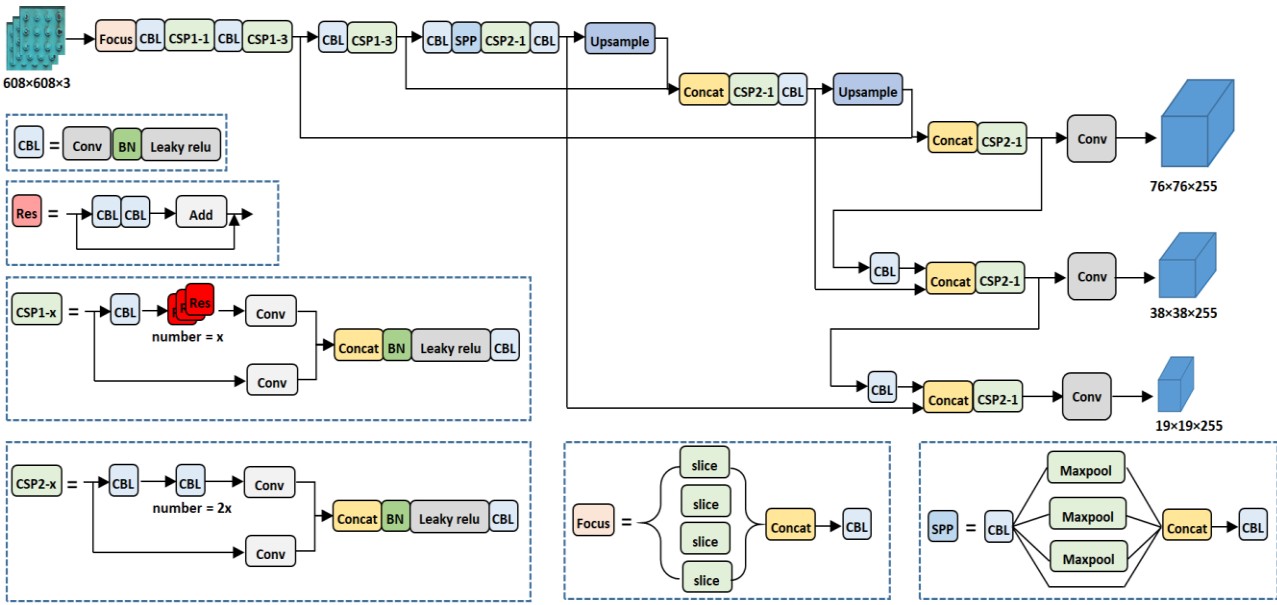

**Figure 8.** Network structure of YOLO v5s.

In the input part, the difference between YOLO v5 and the previous network is the use of mosaic data enhancement and adaptive image scaling. Mosaic data enhancement uses four images randomly, scales them randomly, and then randomly distributes them for splicing. This method enriches the dataset, and the random scaling adds a lot of small targets, making the network more robust. Adaptive image scaling reduces the number of gray pixels supplemented by the short side as much as possible by taking the smallest scaling factor for different image aspect ratios so that the inference speed can be improved.

The difference between YOLO v5 and previous networks in the backbone part is the application of the focus structure and CSP structure, as shown in Figure 8. The feature of the focus structure is the slicing operation. The original $608 \times 608 \times 3$ image is put into the focus structure, in which the slicing operation is used to turn the image into a $304 \times 304 \times 12$ feature map. After a convolution operation with 32 convolution kernels, the image is converted into a feature map of $304 \times 304 \times 32$. The CSP structure divides the feature mapping of the base layer into two parts and then merges them through a cross-stage hierarchical structure, which reduces the amount of calculation while ensuring accuracy.

The neck part is layers arranged between the backbone and the output layer. In the neck part of YOLO v5, the CSP2 structure is adopted to strengthen the ability of network feature integration.

In the prediction part, the feature is the application of a combination of generalized intersection over union (GIOU) loss and distance intersection over union suppression non-maximum suppression (DIOU-NMS), which can obtain higher average precision (AP) than traditional IOU loss and NMS.

The YOLO v5s network has the smallest depth and width of the feature map in the YOLO v5 series. The YOLO v5 series network controls the depth and width of the network by adjusting the depth multiple and width multiple, respectively. For the four models of s, m, l, and x, the depth multiple and width multiples are [0.33, 0.50], [0.67, 0.75], [1.0, 1.0], [1.33, 1.25], and the model size increases accordingly. The increased depth and width enable the model to have higher APs, but the computational efficiency decreases accordingly. Because bolt recognition is a relatively simple detection task compared to multi-type target recognition, even if the APs of the four models should be different in theory, they may not be much different in the bolt dataset. Therefore, the establishment of the bridge bolt recognition dataset, training of these models, and comparing the accuracy of the four models are the main issues to be discussed in this section.

*4.1. Establishment of Bridge Bolt Dataset*

Similar to the previous dataset established for the super-resolution network, the images of the bolt inspection dataset were also taken from three in-service bridges, and 1000 images were selected from these three bridges, then the proposed adaptive scaling unified method was applied to segment these images into 24,000 subimages. Labelimg is a commonly used open-source data labeling software; here, this tool was used to label bolts manually, and a normal bolt was labeled as "bolt", a corroded bolt was labeled as "corrobolt", and a loose bolt was labeled as "loosebolt". After manual labeling, a database of three types of bridge bolts required for training was formed. The results show that the database contained 237,600 normal bolts, 21,583 corroded bolts, and 295 loose bolts. Image examples from the database are shown in Figure 9. The bolts of the small-span suspension bridge were mainly cable clamp bolts and bottom steel truss connecting bolts; the long-span cable-stayed bridge images were mainly the connecting bolts on the side of the steel truss; and the long-span suspension bridge images were cable clamp connecting bolts. In addition, because the collected bolt images were from the images and videos obtained from the multiple inspections of three bridges using UASs, the collected data included bolts in different parts under different lighting conditions. This dataset containing complex conditions and consistent with the real detection scene provided benefits to ensure the robustness of the trained model.

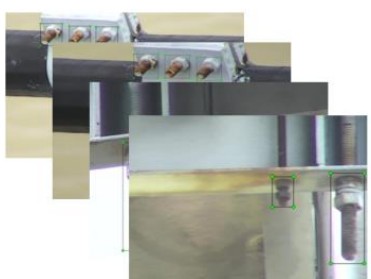 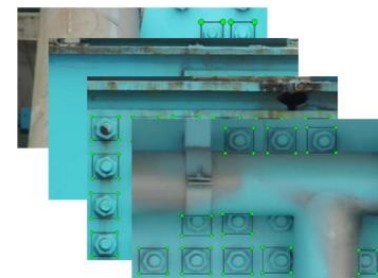 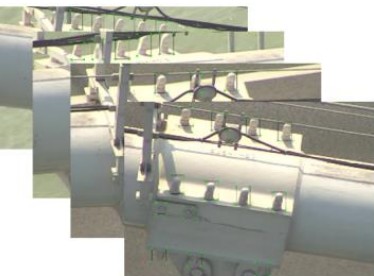

Bolt images from a short span suspension bridge | Bolt images from a long span cable-stayed bridge | Bolt images from a long span suspension bridge

**Figure 9.** Examples of labeled images of the established dataset.

These networks were trained in the PyTorch framework. The computer used for training consisted of an i7 11700k CPU, an Nvidia RTX3090 GPU, and 32G RAM. The number of training steps was 50,000. Before training, the images were divided into training, test, and verification images according to the ratio of 8:1:1. The k-means clustering method was used to calculate the size of 9 anchors, which were [12, 32], [40, 68], [52, 100], [60, 80], [44, 72], [44, 44], [80, 80], [160, 200], and [120, 140].

*4.2. Test Using a Single Object Detection Network*

After training, the mean average precision (mAP)s values of the four models were 0.841, 0.934, 0.936, and 0.940, respectively. The network with the largest network depth and width had the highest accuracy, so YOLO v5x can be selected as the bolt inspection model when real-time inspection is unnecessary. The P-R curves of the four models during the training are shown in Figure 10.

Although the accuracy of YOLO v5x reached 0.940, it can be seen from the P-R curve that the detection accuracy of the three types of bolts was different. After training, the tested classification accuracy of YOLO v5x was 0.883, indicating that some bolts were identified incorrectly. Figure 11 shows a test example of a bolt image after training, where Figure 11a,b are the results of directly identifying bolts and classifying the type of bolts using a single network. The detection results show that the accuracy of bolt classification is 71% in Figure 11a and 88% in Figure 11b, which are the results of the same two images using the object detection network to segment the bolts first and then classify them with a classification network. Figure 11c has an accuracy of 98% and Figure 11d has an accuracy

of 100%. The decimal number on the bounding box of the bolts in the image represents the confidence level of the detection result.

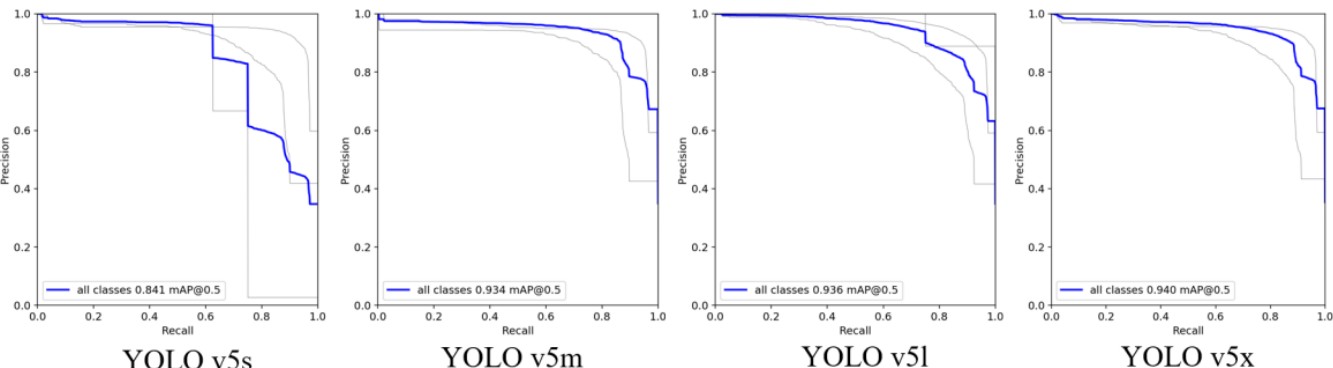

**Figure 10.** P-R curves of the four models during the training.

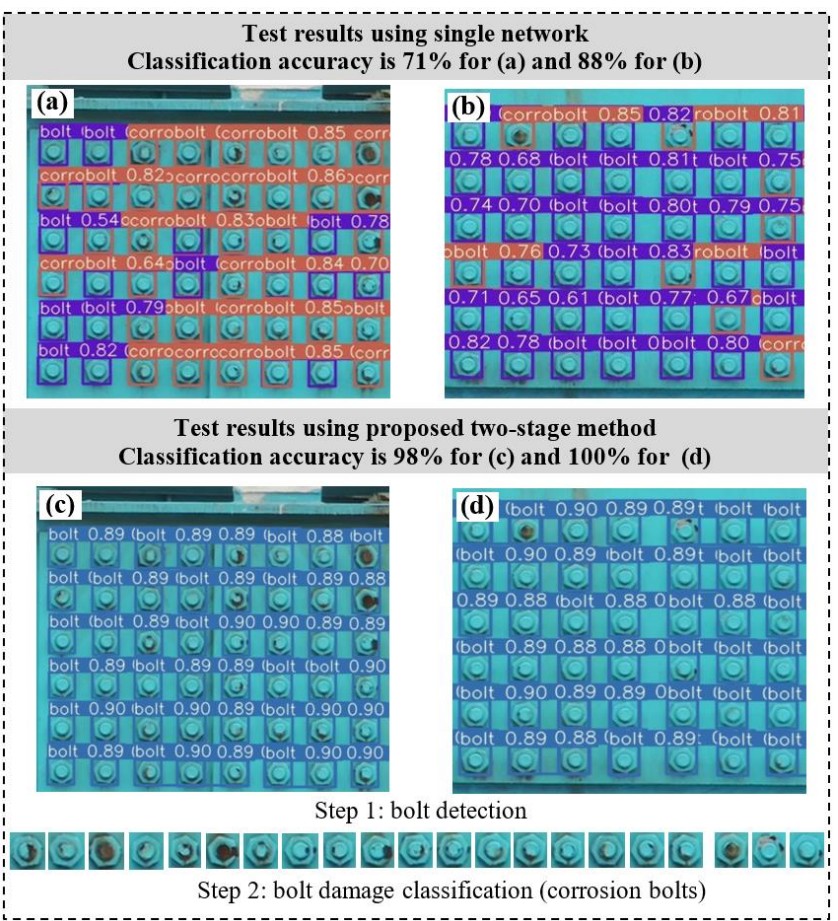

**Figure 11.** Example of test results using two methods: (**a**,**b**) are the results of directly identifying the bolts and classifying the type of bolts using a single network, and (**c**,**d**) are the results of the same two images using an object detection network to first segment the bolts and then classify them with the classification network.

### 4.3. Test Using a Two-Stage Inspection Method

Considering the large number of bolts in bridge inspection, it is difficult to check whether the automatic inspection results are accurate manually. Therefore, the most important goal is to ensure the correct identification of bolt damage. Based on this, a two-stage bolt inspection method using an object detection network and a classification

network was proposed, in which YOLO v5x was used to detect and segment bolts from preprocessed images. Then, efficientNet was applied to classify these segmented bolt images into normal bolts, corrosion bolts, and loose bolts.

Firstly, the bolt database established above was modified by changing the bolts originally labeled as "bolt", "corrobolt", and "loosebolt" into only one category labeled "bolt"; that is, the modified dataset contained only one category of the target. Then, a bolt classification dataset segmented from the three types of bolt datasets in Section 4.1 was assembled to be the training dataset for efficientNet. After that, YOLO v5x was trained again while keeping other parameters consistent, and the mAP was 0.997, which shows that almost all bolts could be detected correctly. The training of efficientNet was also conducted in the PyTorch framework. The number of training steps was 5000, the learning rate was 0.001, and the test accuracy was 0.993 after training, which was much higher than using YOLO v5x trained in Section 4.2. The P-R curve for YOLO v5x and accuracy for efficientNet during training are shown in Figure 12.

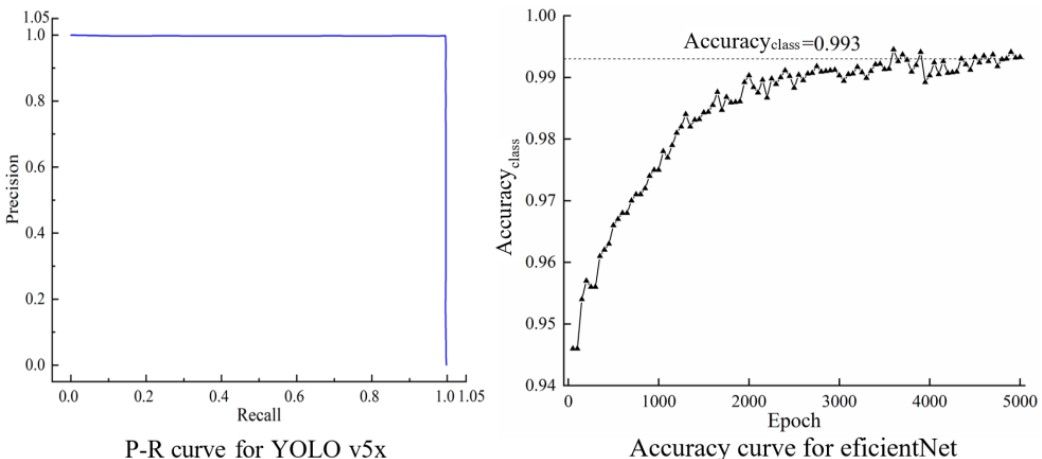

**Figure 12.** The P-R curve and accuracy for YOLO v5x and efficientNet during training.

Figure 11c,d are the results of bolt damage detection using the proposed two-stage method for the same image with Figure 11a,b. The results show that the classification accuracy of bolt damage using the proposed two-stage method was much higher than that using the single network method. The accuracy was 98% for Figure 11c and 100% for Figure 11d. Where the accuracy was calculated by the proportion of the number of bolt damage classes predicted as correct to the total number of bolts for Figure 11a,b, the accuracy was calculated by multiplying the accuracy of bolt identification by the accuracy of bolt damage classification for Figure 11c,d. The accuracy of bolt identification is the ratio of the number of detected bolts to the total number of bolts. The accuracy of bolt classification is the ratio of the number of bolts correctly classified divided by the total number of bolts.

## 5. Filed Test on a Suspension Bridge

An in-service bridge was used as the test object to verify the practicability of the proposed method. This section introduces the test results of the proposed method on a suspension bridge. The inspected bolts were located at the bottom of the bridge and on the cable clamps.

### 5.1. Establishment of Bridge Bolt Dataset

The tested bridge was a pedestrian suspension bridge in the Huaian canal project. The bridge's total length was 197.7 m, the middle span was 115.7 m, and the bridge deck was 2.7 m wide. The bridge deck was a vertical and horizontal channel steel structure with a total of 53 short transverse channels, five longitudinal channels, and 106 cable clamps. The bolts of interest in the test were the channel steel connecting bolts at the bottom of the

bridge and the bolts on the cable clamp, including 371 channel steel connecting bolts and 742 cable clamp bolts. The bridge had been in service for 18 years, and some bolts had been loosened and corroded. However, the bridge was located above the canal, making it difficult to inspect them manually.

The photography strategies of using the UAS to acquire bolt images are described in Section 3.2. For the inspection of cable clamp bolts on the side of the bridge, the UAS hovered at three positions at the ends and the middle of the bridge. The camera was controlled to rotate to take videos of the cable clamp bolts. For the inspection of bolts at the bridge bottom, the UAS hovered at the two ends and the middle of the bottom while the camera took videos from one side to the other. A total of 94 min of bolt video was taken, covering all the bolts that needed to be inspected, including 1113 bolts; 371 bolt images and 2.91 GB of videos were collected. The resolution of the video was 1920 × 1080 pixels, and the frame rate was 30 fps. To keep camera parameters such as exposure and focal length consistent, the shutter speed was set to 1/100 s, and the photo sensibility was set to auto. The focal length remained the same after the start of recording, and the focus was set to autofocus. The filed test is shown in Figure 13.

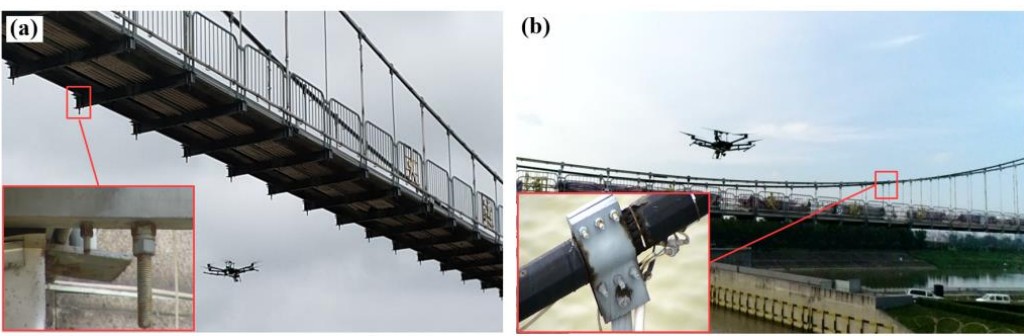

**Figure 13.** Test process of bolt inspection using UAS for (**a**) bridge bottom and (**b**) cable clamp.

*5.2. Data Preprocessing*

After the bolt video was collected, the method above was adopted for step-by-step processing. First, the method based on optical flow and inverse filtering was used to remove the motion blur in the captured video automatically. Since the exposure time of the captured video was the same, the deblurring process of the video did not require manual adjustment of parameters and could be processed automatically. Figure 14 shows examples of preprocessed images using the proposed motion deblurring method and the original images, where Figure 14a,b are motion-blurred images captured from the original video obtained by the UAS, and Figure 14c,d are the two captured images processed by the proposed deblurring method.

After automatically eliminating the video motion blur, the image could be segmented with an adaptive scale. The results calculated using the proposed multi-scale template matching method show that the size of the cable clamp bolts was between 120 × 80 pixels to 300 × 200 pixels, and most of the bolt images from the bridge bottom were between 50 × 100 pixels to 200 × 250 pixels, while the input size of the classification network was 240 × 240 pixels. Most images only needed to be enlarged by two times, so the 2× ESRGAN network was used to process bolt videos in batches. Specifically, for those whose calculated enlarge ratio was greater than 1, 2× super-resolution was then used, then the image would be resized with the ratio of the original enlarge ratio to 2. For bolts whose size was originally larger than 240 × 240 pixels, that is, the calculated enlarge ratio was less than 1, the image was directly resized to that ratio. After unifying image scales, these images were divided according to the size of 640 × 640 pixels with an overlap rate of 10%.

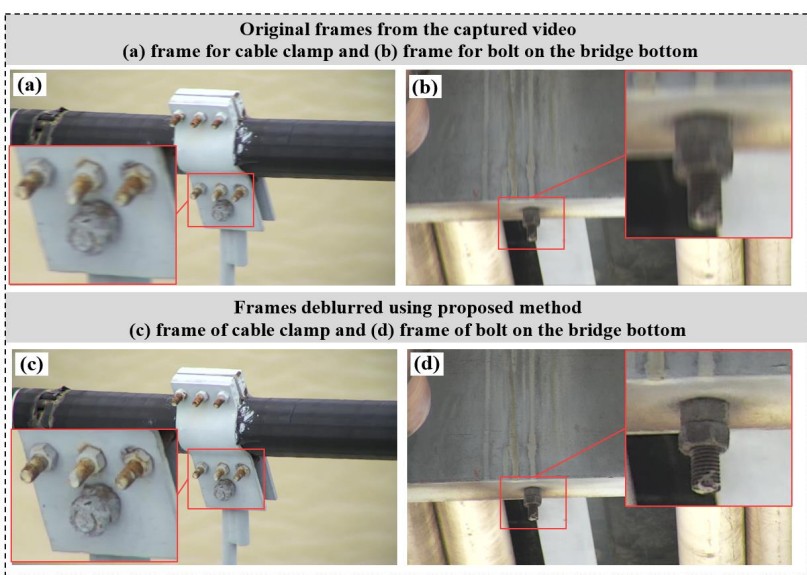

**Figure 14.** Example of original frames and motion deblurred frames: (**a**,**b**) are motion-blurred images captured from the original video obtained by the UAS, and (**c**,**d**) are the two captured images processed by the proposed deblurring method.

*5.3. Bolt Inspection Results*

To verify the accuracy of the proposed two-stage bolt damage automatic identification method, the method of manual inspection and the method of directly using a single network for bolt damage identification that is commonly used in the existing methods were applied as a comparison. Specifically, the proposed method and the existing methods were evaluated for bolt inspection speed and accuracy. The inspection comprised two methods: Method 1 was the proposed method, which included bolt image preprocessing and two-stage bolt damage identification. Method 2 was preprocessing to detect bolt damage by the three categories of YOLO v5x trained in Section 4.2. Method 3 was using the technique of existing research, which is using only the YOLO v5x network for bolt detection and classification. The test objects were 100 images derived from the acquired video. Among them, 50 images contained motion blur. The bolts in the images include both cable clamp bolts and bridge bottom bolts. Table 2 shows the computing time required for the image preprocessing method, the two-stage inspection method, and the single-stage identification method to analyze 100 images. The computer used for data processing was the same as that for model training. The results show that the computing time of the proposed method was much longer than that of the single-stage identification method, which is the result of the sacrifice of inference speed in the tradeoff between speed and accuracy and the disadvantage of the proposed method. To make up for this shortcoming as much as possible, the proposed method was programmed into the onboard computer of the UAS so that automatic analysis could run continuously during flight. This can reduce the total time of data acquisition and analysis. Since the i7 10710U CPU of the onboard computer provided satisfactory computing ability, the processing time of bolt images using the proposed method was tested on it, and the time for deblurring was 834.2 s/100 images, while the time for uniform scaling was 1162.5 s/100 images. In addition, considering that the focus of this study was to achieve the highest accuracy possible, there was no optimization in the algorithm's efficiency at present, and the efficiency of the algorithm will be improved in future work.

Table 3 shows the accuracy of 100 images calculated using the above three methods: TP represents the true positive, which is the number of bolts that should be detected as positive types that were finally detected as positive in the test. TN represents the true negative, which is the number of bolts that should be detected as negative types that were detected as negative. Both TP and TN mean that the bolts were inspected correctly. FP

is the false positive, and FN is the false negative. They mean that bolts were incorrectly detected as categories in which they did not belong. The calculation method of accuracy is:

$$\text{accuracy}_{\text{total}} = \frac{\text{TP} + \text{TN}}{\text{TP} + \text{TN} + \text{FP} + \text{FN}} \tag{14}$$

**Table 2.** Bolt image processing time of the three methods.

| Method | | Time | |
|---|---|---|---|
| Proposed method (image preprocessing and two-stage inspection) | Step1: Image preprocessing | Deblurring Uniform scale | 611.6 s 592.7 s |
| | Step 2: Two-stage bolt inspection | 11.5 s | |
| Traditional method | One-stage bolt inspection | 6.9 s | |

**Table 3.** Inspection accuracy using the three methods.

| Method | Group | TP | TN | FP | FN | Accuracy |
|---|---|---|---|---|---|---|
| 1. Proposed method (preprocessing and two-stage method) | Blurred | 102 | 36 | 2 | 0 | 0.986 |
| | Non-blurred | 169 | 96 | 1 | 0 | 0.996 |
| 2. Preprocessing and single network | Blurred | 91 | 25 | 15 | 9 | 0.829 |
| | Non-blurred | 150 | 86 | 19 | 11 | 0.884 |
| 3. Using single network | Blurred | 79 | 18 | 29 | 22 | 0.655 |
| | Non-blurred | 143 | 79 | 27 | 17 | 0.835 |

The results in Table 3 show that the proposed method had the best accuracy for both images with and without motion blur. The detection accuracy of method 2 for images with and without motion blur was similar but lower than that of the proposed method. Method 3 not only had the lowest accuracy but also had worse accuracy for motion-blurred images. This proves that the presented bolt image preprocessing method was helpful in making the detection accuracy stable, and the proposed two-stage inspection method had higher accuracy than using a single network. Specifically, the comparison of the accuracy with and without motion blur in method 1 shows that the proposed image preprocessing method could effectively eliminate the influence of image blur on the accuracy, while the comparison of the accuracy with and without motion blur in method 3 shows that motion blur led to a decrease in accuracy of 0.18. The comparison of the accuracy between method 1 and method 2 proved that the proposed two-stage inspection method had better accuracy than the single network method. The average accuracy was improved by 0.13. In general, the primary goal of bolt inspection in this research was to make the detection accuracy as close to 100% as possible so as to ensure that there were as few missing identifications of bolt damage as possible. The accuracy of the proposed methods was higher than 0.98, which can satisfy engineering requirements.

The proposed bolt image preprocessing method involves two parts: image deblurring and adaptive scale segmentation. Considering that both parts required a long computing time, the contribution of the two parts to the improvement of accuracy was tested respectively. Three methods were established for the comparison. Method A: remove the motion blur of the image first and then use the two-stage inspection method for detection. Method B: carry out adaptive segmentation of the image first and then use the two-stage inspection method for detection. Method C: only use the two-stage inspection method for detection. Because the number of pixels occupied by bolts in this field test was large, and the advantage of the adaptive scale segmentation method may be more evident for images with small bolts, the dataset here was assembled with 50 motion-blurred images of this field test and 50 motion-blurred images from a steel truss bridge, as shown in the middle of Figure 9. The test results show that the average accuracy of method A was 0.941, in which the average accuracy of 50 images with large bolt targets was 0.979, and the average accuracy of 50 images with small bolt targets was 0.903. The accuracy of method B was 0.899, in which the average accuracy was 0.901 for 50 images with large bolt targets and

0.897 for 50 images with small bolt targets. The accuracy of method C was 0.795, in which the average accuracy was 0.885 for 50 images with large bolt targets and 0.705 for 50 images with small bolt targets. The accuracy of method A was 0.146 higher than that of method C, while compared with method C, the accuracy of method B was improved by 0.104, which indicates that the motion deblurring contributed more to the improvement of accuracy than adaptive scale segmentation. Among these results, the accuracy for images with large bolt sizes was 0.076 more than for images with small bolt sizes in method A, while this accuracy differed little in method B, which proves that adaptive scale segmentation was more effective in improving the accuracy when the bolt scale in the image was small. Based on the above results, a better combination of bolt inspection methods can be selected according to different scenarios. For the inspection of bolts that occupy large pixels in the image, such as the inspection of cable clamp bolts, it is better to perform motion deblurring first and then inspect using the two-stage inspection method. For the inspection of bolts whose size in the image is small, such as the inspection of nodes of a steel truss bridge, the combination of motion deblurring and adaptive scale segmentation should be used for preprocessing, and then using the two-stage inspection method.

Figure 15 shows the bolt health condition of the inspected bridge and examples of loose bolts and corrosion bolts. It shows that nearly 9.1% of the bridge bottom bolts had been loosened, and most of them were corroded to a certain extent. The test results shown in Figure 15 provide intuitive and practical guidance for the health management of the bridge and prove that the proposed method is suitable for in-service bridges.

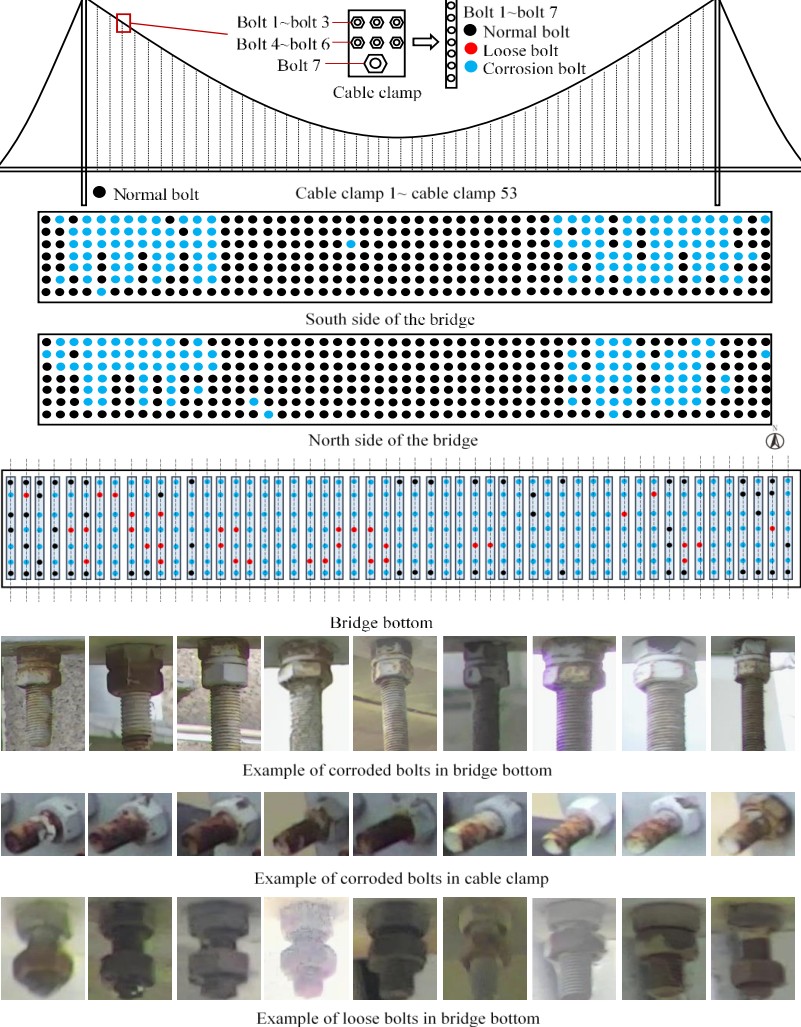

**Figure 15.** Bolt inspection result of the suspension bridge.

## 6. Conclusions

This study proposes a practical method for the inspection of bridge bolts using UASs and deep learning. The specific conclusions are as follows:

1. The proposed motion deblur method that applies inverse filtering to eliminate the motion blur of the captured image automatically with the estimated movement direction and displacement by optical flow was tested. The results showed that it effectively removed motion blur caused by the moving and rotating of the UAS.

2. An adaptive scale unified segmentation method based on multi-scale template matching and super-resolution was proposed to address the low classification accuracy problem for images with small bolt sizes, in which ESRGAN was trained and applied to up-sample bolt images to the required size with detailed texture. The comparison between the trained ESRGAN, VDSR, and bicubic interpolation showed that images enlarged by ESRGAN had the highest fidelity.

3. The proposed two-stage bolt damage identification used the YOLO v5x network to segment bolts from preprocessed images and then classify them into normal bolts, corrosion bolts, and loose bolts using efficientNet. The trained network showed 99.7% accuracy on bolt detection and 99.3% accuracy on bolt damage classification in our dataset containing 24,000 images, which was much higher than using one network for both bolt detection and bolt damage classification.

4. The proposed method was verified on an in-service suspension bridge. The results showed that the proposed motion deblurring method showed improvement of 14.6% in bolt damage detection accuracy with motion-blurred images, and the adaptive scale unified segmentation method showed 10.4% greater accuracy on images with small bolt sizes.

As described in Table 2, the proposed method selects the direction to improve the accuracy as much as possible in the tradeoff between accuracy and speed, resulting in the proposed method being far from realizing real-time analysis. Future work will focus on researching the lightweight bolt image preprocessing method to reduce the time consumed in processing as much as possible. In addition, the automatic inspection path planning of the UAS is also meaningful work. Based on the initial three-dimensional model of the bridge and path planning algorithm, the optimal inspection path of the UAS can be automatically planned. On the one hand, the control of the UAS can eliminate the dependence on manual operations and realize whole process automation from data acquisition to analysis. On the other hand, a reasonably planned path and flight speed can increase flight efficiency and improve the quality of collected data.

**Author Contributions:** S.J.: conceptualization, methodology, algorithm, experiment, validation, writing original draft, writing—review and editing. J.Z.: conceptualization, methodology, validation, writing—review and editing. W.W.: experiment, validation. Y.W.: experiment, validation, writing—review and editing. All authors have read and agreed to the published version of the manuscript.

**Funding:** The research presented was financially supported by the Key R&D Program of Jiangsu (No.: BE2020094), the National Key R&D Program of China (No.: 2019YFC1511105, No.:2020YFC1511900, and No.:2022YFC3801700).

**Data Availability Statement:** A part of the dataset established in the study and the trained model can be downloaded from the link: https://github.com/shark-J/bridge-bolt-inspection (accessed on 6 November 2022).

**Conflicts of Interest:** The authors declare no conflict of interest.

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
