# Peer review of "Automatic Inspection of Bridge Bolts Using Unmanned Aerial Vision and Adaptive Scale Unification-Based Deep Learning"

_remotesensing, doi:10.3390/rs15020328_

Round 1

Reviewer 1 Report

The manuscript developed a vision-based method using deep learning and un- manned aerial vision to inspect the bolt's condition. The subject is very important in bridge maintenance and the approach is novel. The authors are suggested to polish the language of the paper before its publication.

Author Response

We gratefully thank the reviewer for the valuable comments and suggestions. We have carefully considered all comments from the reviewer and revised our manuscript accordingly. The manuscript has been double-checked, and the typos and grammar errors have been corrected. The language presentation was also improved with assistance from a native English speaker with an appropriate research background.

Reviewer 2 Report

Dear Authors,

The scientific paper presents a model for automatic bridge inspection. The presented model for bridge inspection is based on unmanned aerial vision and adaptive scale unification-based deep learning. This method is a big step forward compared to the existing visual method of inspecting bridges. The authors chose to test the method on bolts as bridge parts. The proposed method is verified on bridge to detect bolts and classify them into normal bolts, corrosion bolts, and loose bolts. This kind of detection method uses a camera to collect bolt images rapidly and uses analysis methods based on machine vision to automatically identify bolts and classify whether there are defects in bolts. This method contains three parts: bolt image acquisition, data preprocessing, and bolt damage identification. Scientific work has quality. The literature review was done satisfactorily, enumerating the various methods based on deep learning that have been used so far for bridge inspection. Methods for collecting images using cameras are also listed in the paper, which is a significant improvement compared to traditional methods. Although previous research also used a camera to collect images, in this work, special attention is paid to the quality of the images and their blurriness due to the high speed of unmanned aerial vehicles. The collected data and methods of image processing, how to obtain the best possible quality, were presented by the authors in great detail. The method of collecting data on bolts was based on two steps, where in the first step the screws were detected, and in the second, their classification based on defects was carried out. The methodology has been verified on a practical example and has satisfactory accuracy, which is proven by means of numerical values.

The recommendations for correcting the scientific paper are as follows

·         Line 19- space missing “them into sub-images with uniform bolt size. (2)To address the problem that directly applying an”.

·         Line 77- Is it necessary to add explained terms “yolov4 FPM”?

·         Line 101 – What is UAV? It is mentioned several times in the text but not explained.

·         Line 222 – space missing “of ???(??,??)can be”.

·         Line 341 – By - lowercase letter.

·         Line 370 – “results” instead “resluts”.

·         Line 453- What is Labelimg?

·         Could you please adjust the text so that the figures are mentioned first in the text and then they appear? I think that way it will be easier to follow the text. This is particularly noticeable when referring to Figures 7 and 13.

·         Line 486- Please explain and write what parts a, b, c, d represent in Figure 11. Write both below the figure and in the text.

·         Line 534- period (.) at the end of a sentence - “section 3.2, For the inspection”.

·         Line 535 - period (.) at the end of a sentence – “bridge, The camera was”.

·         Line 553- Please explain and write what parts a, b, c, d represent in Figure 14. Write both below the figure and in the text.

·         Line 595- Please explain in more detail what represent:  TP (true positive), TN (true negative), FP (false positive), and FN (false negative).

·         Please put spaces between numerical values ​​and units of measure throughout the text (for example “872 mm” instead “872mm”).

·         I noticed that the authors cited themselves 6 times out of a total of 39 references (References no. 16,17,18,19, 30, 32).

Author Response

Response to Reviewer 2 Comments

General evaluation: The scientific paper presents a model for automatic bridge inspection. The presented model for bridge inspection is based on unmanned aerial vision and adaptive scale unification-based deep learning. This method is a big step forward compared to the existing visual method of inspecting bridges. The authors chose to test the method on bolts as bridge parts. The proposed method is verified on bridge to detect bolts and classify them into normal bolts, corrosion bolts, and loose bolts. This kind of detection method uses a camera to collect bolt images rapidly and uses analysis methods based on machine vision to automatically identify bolts and classify whether there are defects in bolts. This method contains three parts: bolt image acquisition, data preprocessing, and bolt damage identification. Scientific work has quality. The literature review was done satisfactorily, enumerating the various methods based on deep learning that have been used so far for bridge inspection. Methods for collecting images using cameras are also listed in the paper, which is a significant improvement compared to traditional methods. Although previous research also used a camera to collect images, in this work, special attention is paid to the quality of the images and their blurriness due to the high speed of unmanned aerial vehicles. The collected data and methods of image processing, how to obtain the best possible quality, were presented by the authors in great detail. The method of collecting data on bolts was based on two steps, where in the first step the screws were detected, and in the second, their classification based on defects was carried out. The methodology has been verified on a practical example and has satisfactory accuracy, which is proven by means of numerical values.

Response: We gratefully thank the reviewer for the valuable comments and suggestions. We have carefully considered all comments from the reviewer and revised our manuscript accordingly. The manuscript has been double-checked, and the typos and grammar errors have been corrected. The language presentation was also improved with assistance from a native English speaker with an appropriate research background. The following is a discussion of our response.

Point 1: Line 19- space missing "them into sub-images with uniform bolt size. (2)To address the problem that directly applying an".

Response 1: Thanks for the careful check. The mentioned part has been modified.

Point 2: Line 77- Is it necessary to add explained terms "yolov4 FPM"?

Response 2: Thanks for the review work. The yolov4 FPM is an improved network by the authors in reference 13. As a reference, we believe spending space explaining the FPM is unnecessary. Here, we modify the yolov4 FPM to "an improved YOLO v4 network".

Point 3: Line 101 – What is UAV? It is mentioned several times in the text but not explained.

Response 3: Thanks for the careful check, the UAV and UAS in the paper represent unmanned aerial vehicles/systems. They have been unified into UAS to avoid ambiguity.

Point 4: Line 222 – space missing "of ???(??,??)can be".

Response 4: Thanks for the careful check. The mentioned part has been modified.

Point 5: Line 341 – By - lowercase letter.

Response 5: Thanks for the careful check. The mentioned part has been modified.

Point 6: Line 370 – "results" instead "resluts".

Response 6: Thanks for the careful check. The mentioned part has been modified.

Point 7: Line 453- What is Labelimg?

Response 7: Labelimg is a commonly used open-source data labeling software. The explanation of this tool has been added to the paper.

Point 8: Could you please adjust the text so that the figures are mentioned first in the text and then they appear? I think that way it will be easier to follow the text. This is particularly noticeable when referring to Figures 7 and 13.

Response 8: Thanks for the review work, the position of the figures in the paper has been repositioned to make it easier to read.

Point 9: Line 486- Please explain and write what parts a, b, c, d represent in Figure 11. Write both below the figure and in the text.

Response 9: Thanks for the review work, the explanation of a, b, c, d has been added to the text and the title of figure 11.

Detailed: (page 15)

Figure 11 shows a test example of the bolt image after training, where (a) and (b) are the results of directly identifying bolts and classifying the type of bolts using a single network. The detection results show that the accuracy of bolt classification is 71% for (a) and 88% for (b). (c) and (d) are the results of the same two images using the object detection network to segment the bolts first and then classify them with a classification network, where (c) has an accuracy of 98% and (d) has an accuracy of 100%. The decimal number on the bounding box of the bolts in the image represents the confidence level of the detection result.

Point 10: Line 534- period (.) at the end of a sentence - "section 3.2, For the inspection".

Response 10: Thanks for the careful check, the mentioned part has been modified.

Point 11: Line 535 - period (.) at the end of a sentence – "bridge, The camera was".

Response 11: Thanks for the careful check, the mentioned part has been modified.

Point 12: Line 553- Please explain and write what parts a, b, c, d represent in Figure 14. Write both below the figure and in the text.

Response 12: Thanks for the review work, the explanation of a, b, c, d has been added to the text and the title of figure 14.

Detailed: (page 17)

Figure 14 shows examples of preprocessed images using the proposed motion deblurring method and the original images, where (a) and (b) are motion-blurred images captured from the original video obtained by the UAS, and (c) and (d) are the two captured images processed by the proposed deblurring method.

Point 13: Line 595- Please explain in more detail what represent:  TP (true positive), TN (true negative), FP (false positive), and FN (false negative).

Response 13: Thanks for the review work, detailed explanations of these definitions are added to the paper.

Detailed: (page 19)

TP represents the true positive, which is the number of bolts that should be detected as positive types that were finally detected as positive in the test. TN represents the true neg-ative, which is the number of bolts that should be detected as negative types were detected as negative. Both TP and TN mean that the bolts were inspected correctly. FP is the false positive, and FN is the false negative. They mean that bolts were incorrectly detected as categories that did not belong to them.

Point 14: Please put spaces between numerical values and units of measure throughout the text (for example "872 mm" instead "872mm").

Response 14: Thanks for the careful check, the mentioned part has been modified.

Point 15: I noticed that the authors cited themselves 6 times out of a total of 39 references (References no. 16,17,18,19, 30, 32).

Response 15: Thanks for the review work. We re-examined the cited author's research and deleted the research (references 16 and 19) that is not closely related to this paper.

Reviewer 3 Report

The authors developed an image-based bridge bolt inspection system. The system consists of an UAV system for image data collection, a motion deblurring method for enhancing image quality, an adaptive scale unified segmentation method for unifying the image size, and a two-stage bolt damage identification method to determine corroded or loosen bolts. The present study collected and analyzed a large sample dataset to develop and test the proposed methods. The manuscript is well organized and appears to fit the overall scope of Remote Sensing. However, before it can be accepted, there are some doubts that should be cleared.

1. Section 4.1. How do the authors define loosened bolts? Is this based on the bolt head rotation angle? If yes, how many angels shall be considered as loosened? It this based on the height change of bolt head, or based on something else?

2. The results show that the preprocessing algorithm proposed by the author can improve the detection accuracy, but the optical flow needed for motion debarring, and the ESGRAN used in the super-resolution both involve heavy amounts of computation, this is also verified in the results of Table 2 presented in the paper. The author needs to focus on the time-consuming problem of the algorithm in the conclusion and future work.

3. The authors argued that existing methods were affected by background and illumination. It is unclear if the proposed method by the authors is also affected by illumination. It is suggested that the author give a detailed description of the robustness of the proposed method.

4. Many consumer-grade DJI drones or similar drone products can shortly freeze the camera position when collecting images. Do the authors investigate anything in this aspect? Whether this function can reduce the motion blur of the captured bolt image?

5. Is the ‘accuracy’ in Figure 12 and Equation 13 the same parameter? If not, please use different names to distinguish.

6. Is it necessary to emphasize the equation in Figure 4? If not, it is recommended to put it in the text.

7. Does UAV and UAS stand for the same thing in the paper? If so, it is recommended to unify them into one.

8. Page 9, line 322. Not all readers are familiar with the definition of ‘nonlinear mapping’, the related references should be cited.

9. Page 11, line 363. The parameter mean square error (MSE) should be given the related references, such as: https://doi.org/10.1007/s13349-022-00635-8

10. Page 11, line 369. The ‘Gaussian function’ should be provided references,

such as: https://doi.org/10.1177/13694332221126374

11. Figure 11: The figure needs to be explained more in Sections 4.2 and 3. In particular, the decimal in the image needs to be elaborated in the text.

12. The ‘2x, 4x, and 8x’ in the paper used ‘x’ instead of ‘×’, these parts should be checked and revised.

13. Page 14, line 475. What is the ‘mAPs’?

Author Response

Response to Reviewer 3 Comments

General evaluation: The authors developed an image-based bridge bolt inspection system. The system consists of an UAV system for image data collection, a motion deblurring method for enhancing image quality, an adaptive scale unified segmentation method for unifying the image size, and a two-stage bolt damage identification method to determine corroded or loosen bolts. The present study collected and analyzed a large sample dataset to develop and test the proposed methods. The manuscript is well organized and appears to fit the overall scope of Remote Sensing. However, before it can be accepted, there are some doubts that should be cleared.

Response: We gratefully thank the reviewer for the valuable comments and suggestions. We have carefully considered all comments from the reviewer and revised our manuscript accordingly. The manuscript has been double-checked, and the typos and grammar errors have been corrected. The language presentation was also improved with assistance from a native English speaker with an appropriate research background. The following is a discussion of our response.

Point 1: Section 4.1. How do the authors define loosened bolts? Is this based on the bolt head rotation angle? If yes, how many angels shall be considered as loosened? It this based on the height change of bolt head, or based on something else?

Response 1: Thanks for the review work. The method of judging bolt looseness is based on the manual visual judgment of whether the nut moves outward or falls off, this judgment method is also a commonly used method for studies that use machine vision to detect structural damages. We have noticed that some studies have used the rotation angle of the nut to judge whether it is loose. The premise of this method is that the initial position of the bolt is marked on the steel when the installation is completed, but we rarely find such mark lines in the inspection of serval bridges, including the Hu-Su-Tong Bridge, which is the first highway and railway bridge with a span of more than 1000 meters in the world, and its bolts have almost no marking lines. The purpose of our proposed method is to qualitatively detect bolt defects and replace manual daily visual inspection. This study proves that the method based on deep learning can achieve similar accuracy to manual judgment when processing bolt images. Therefore, we think the proposed method is meaningful.

Point 2: The results show that the preprocessing algorithm proposed by the author can improve the detection accuracy, but the optical flow needed for motion debarring, and the ESGRAN used in the super-resolution both involve heavy amounts of computation, this is also verified in the results of Table 2 presented in the paper. The author needs to focus on the time-consuming problem of the algorithm in the conclusion and future work.

Response 2: We agree with the reviewers that real-time inspection is absolutely important for applying UAS for structural damage detection (cracks, concrete spalling, etc.). However, the difference between the bolt inspection and the above damages is that the cracks, spalling mainly harm the durability of the structure, but once the bolts are loose or missing, they will directly endanger the safety of the structure. In particular, we found that two or more bolts of the main cable clamp of some long-span suspension bridges loosed or missed during the inspection, since a cable clamp is fixed by about a dozen bolts, as shown in the figure below, this has greatly attracted the attention of the bridge management department. Similarly, for steel truss bridges, the direct connection between components completely depends on the anchoring of bolts. Once the bolts are lost or missing, it means that part of the anchoring force disappears. Therefore, the requirement of the bridge management department we cooperate with is to ensure that all bolt damages can be correctly identified as far as possible. As described in the manuscript, we put the accuracy of bolt inspection in the first place.

On the other hand, our experience tells us that real-time requirement for bolt inspection is lower than those of cracks, spalling. For crack detection, if real-time cannot be achieved, a key problem is not knowing how much area the UAS has inspected. In this case, off-line detection may lead to a missed inspection. The difference in bolt detection is that the bolts are only distributed at specific positions on the bridge surface, like cable clamp or bridge side joints. Therefore, even if there are positions where bolts are not completely photographed, these positions can be quickly found for supplementary photographing.

In a word, in terms of the balance between inspection accuracy and speed, the idea of this study is to consider the particularity of bolt detection and put forward a method to improve inspection accuracy. The accuracy higher than 98% on the test is also satisfactory. In order to make up for the time-consuming problem of the algorithm as much as possible, we applied the onboard computer of the UAS to provide continuous computing ability during the inspection.

Point 3: The authors argued that existing methods were affected by background and illumination. It is unclear if the proposed method by the authors is also affected by illumination. It is suggested that the author give a detailed description of the robustness of the proposed method.

Response 3: Thanks for the review work. In fact, compared with the traditional image processing methods, one of the obvious advantages of the inspection method based on deep learning is that it is not sensitive to light interference. It is generally believed that as long as there are images with complex backgrounds and light changes in the training set, when the model is correctly trained, the model will also have a good ability to distinguish these interference factors. Because the images in the bolt dataset established in this paper are from the detection images of multiple in-service bridges, the established dataset has included the bolt images taken in a variety of background conditions and obvious light changes, so the proposed method also has the ability to resist the interference of environmental light to a certain extent. On the other hand, when inspecting a bridge, the weather with good illumination will be chosen before inspection, it is the basis of visual inspection. Therefore, there will be no extremely poor illumination environment during the inspection. Discussion about light-sensitive has been added to the paper.

Detailed: (page 14)

The bolts of the small-span suspension bridge are mainly the cable clamp bolts and the bottom steel truss connecting bolts, the long-span cable-stayed bridges are mainly the connecting bolts on the side of the steel truss, and the long-span suspension bridges are the cable clamp connecting bolts. In addition, because the collected bolt images are from the images and videos obtained from the multiple inspections of three bridges using UAS, the collected data includes bolts in different parts under different lighting conditions. This dataset containing complex conditions and consistent with the real detection scene pro-vides benefits to ensure the robustness of the trained model.

Point 4: Many consumer-grade DJI drones or similar drone products can shortly freeze the camera position when collecting images. Do the authors investigate anything in this aspect? Whether this function can reduce the motion blur of the captured bolt image?

Response 4: Thanks for the review work. As mentioned by the reviewer, we know that some commercial UAVs have the function of locking the camera to take photos. In fact, the UAV and camera used in this paper are from DJI company, so it also has the mentioned function. This function does play a role in shooting objects with a large viewing angle, but when using a telephoto camera, this function will not bring benefits, and even make the camera suddenly change its direction to shoot surrounding objects instead of bolts. When shooting with a telephoto camera, the most important thing is to make the camera point to the shooting position stably, because any small jitter will lead to sharp changes in the picture. Therefore, when acquiring video, we usually set the camera to "free mode", that is, the camera does not rotate due to the rotation of the UAV itself. However, the function mentioned by the reviewer is to lock the gimbal of the camera at the moment of shooting. This instantaneous locking will generally lead to a slight rotation of the camera, which is fatal to the telephoto camera. In addition, this function is only effective for taking photos, while this paper adopts the way of taking videos. See comment 3 for relevant explanations.

Point 5: Is the ‘accuracy’ in Figure 12 and Equation 13 the same parameter? If not, please use different names to distinguish.

Response 5: Thanks for the careful check, the accuracy in these two places represents different meanings. The accuracy in Figure 12 represents the change of the accuracy of efficientNet during the training process, which represents the accuracy of the segmented bolts in classifying to the three defects. And the accuracy represented by Equation 13 is the accuracy of the whole set of methods to inspect the bolts, which contains the accuracy of first segmenting the bolts from the original image and the accuracy of the bolts being correctly classified. To distinguish the two accuracies, the accuracy in Figure 12 is modified to accuracyclass and the one in Equation 13 is modified to accuracytotal.

Point 6: Is it necessary to emphasize the equation in Figure 4? If not, it is recommended to put it in the text.

Response 6: Thanks for the reviewer's comments, the equation in Figure 4 have been moved to the text.

Detailed: (page 8)

The inverse filtering method was used with the calculated blur kernel size and direction to remove the motion blur directly, it is calculated as shown in Equation 8, where T is the exposure time, a and b are the horizontal and vertical movement. The steps of the process are shown in Figure 4.

Point 7: Does UAV and UAS stand for the same thing in the paper? If so, it is recommended to unify them into one.

Response 7: Thanks for the careful check, the UAV and UAS in the paper represent unmanned aerial vehicles/systems, they have been unified into UAS according to the suggestions of reviewers.

Point 8: Page 9, line 322. Not all readers are familiar with the definition of ‘nonlinear mapping’, the related references should be cited.

Response 8: Thanks for the reviewer's comments, the related reference has been added to the paper.

  1. Zeng, X., & Huang, H. (2012). Super-resolution method for multiview face recognition from a single image per per-son using nonlinear mappings on coherent features. IEEE signal processing letters, 19(4), 195-198, https://doi.org/10.1109/LSP.2012.2186961

Point 9: Page 11, line 363. The parameter mean square error (MSE) should be given the related references, such as: https://doi.org/10.1007/s13349-022-00635-8

Response 9: Thanks for the reviewer's comments, the suggested reference has been added to the paper.

  1. Zhao, H. W., Ding, Y. L., Li, A. Q., Chen, B., & Wang, K. P. (2022). Digital modeling approach of distributional map-ping from structural temperature field to temperature-induced strain field for bridges. Journal of Civil Structural Health Monitoring, 1-17, https://doi.org/10.1007/s13349-022-00635-8

Point 10: Page 11, line 369. The ‘Gaussian function’ should be provided references, such as: https://doi.org/10.1177/13694332221126374

Response 10: Thanks for the reviewer's comments, the suggested reference has been added to the paper.

  1. Lin, S. W., Du, Y. L., Yi, T. H., & Yang, D. H. (2022). Influence lines-based model updating of suspension bridges considering boundary conditions. Advances in Structural Engineering, 13694332221126374, https://doi.org/10.1177/13694332221126374

Point 11: Figure 11: The figure needs to be explained more in Sections 4.2 and 3. In particular, the decimal in the image needs to be elaborated in the text.

Response 11: Thanks for the review work, the explanation of figure 11 has been added to the text and the title of figure 11.

Detailed: (page 15)

Figure 11 shows a test example of the bolt image after training, where (a) and (b) are the results of directly identifying bolts and classifying the type of bolts using a single network. The detection results show that the accuracy of bolt classification is 71% for (a) and 88% for (b). (c) and (d) are the results of the same two images using the object detection network to segment the bolts first and then classify them with a classification network, where (c) has an accuracy of 98% and (d) has an accuracy of 100%. The decimal number on the bounding box of the bolts in the image represents the confidence level of the detection result.

Point 12: The ‘2x, 4x, and 8x’ in the paper used ‘x’ instead of ‘×’, these parts should be checked and revised.

Response 12: Thanks for the careful check, these part has been modified.

Point 13: Page 14, line 475. What is the ‘mAPs’?

Response 13: Thanks for the reviewer's comments, mAP refers to mean average precision, its full name has been added to the paper.
